# Thermal Expansion and Magnetostriction of Laves-Phase Alloys: Fingerprints of Ferrimagnetic Phase Transitions

**DOI:** 10.3390/ma12111755

**Published:** 2019-05-30

**Authors:** Chao Zhou, Huixin Bao, Yoshitaka Matsushita, Tieyan Chang, Kaiyun Chen, Yin Zhang, Fanghua Tian, Wenliang Zuo, Xiaoping Song, Sen Yang, Yang Ren, Xiaobing Ren

**Affiliations:** 1School of Science, MOE Key Laboratory for Nonequilibrium Synthesis and Modulation of Condensed Matter, State Key Laboratory for Mechanical Behavior of Materials, Xi’an Jiaotong University, Xi’an 710049, China; chao.zhou@xjtu.edu.cn (C.Z.); baohx2006@gmail.com (H.B.); changty.6@stu.xjtu.edu.cn (T.C.); yzhang18@xjtu.edu.cn (Y.Z.); tfh2017@xjtu.edu.cn (F.T.); zuowenliang@xjtu.edu.cn (W.Z.); songxp@mail.xjtu.edu.cn (X.S.); 2National Institute for Materials Science, Beamline BL15XU, Spring-8, 1-1-1 Kohto, Sayo-cho, Hyogo 679-5148, Japan; Matsushita.Yoshitaka@nims.go.jp; 3Department of Mechanical Engineering, The University of Melbourne, Parkville, Victoria 3010, Australia; chenkaiyun721@stu.xjtu.edu.cn; 4X-Ray Science Division, Advanced Photon Source, Argonne National Laboratory, Argonne, IL 60439, USA; ren@aps.anl.gov; 5Center for Functional Materials, National Institute for Materials Science, Tsukuba, Ibaraki 305-0047, Japan

**Keywords:** magnetostriction, thermal expansion, ferrimagnetic transition, Laves-phase alloy

## Abstract

The magneto–elastic coupling effect correlates to the changes of moment and lattice upon magnetic phase transition. Here, we report that, in the pseudo-binary Laves-phase Tb_1*-x*_Dy*_x_*Co_2_ system (*x* = 0.0, 0.7, and 1.0), thermal expansion and magnetostriction can probe the ferrimagnetic transitions from cubic to rhombohedral phase (in TbCo_2_), from cubic to tetragonal phase (in DyCo_2_), and from cubic to rhombohedral then to tetragonal phase (in Tb_0_*_._*_3_Dy_0.7_Co_2_). Furthermore, a Landau polynomial approach is employed to qualitatively investigate the thermal expansion upon the paramagnetic (cubic) to ferrimagnetic (rhombohedral or tetragonal) transition, and the calculated thermal expansion curves agree with the experimental curves. Our work illustrates the correlation between crystal symmetry, magnetostriction, and thermal expansion in ferrimagnetic Laves-phase alloys and provides a new perspective to investigate ferrimagnetic transitions.

## 1. Introduction

Magnetic transition is the physical fundamental for magnetism and magnetic materials [1,2]. Thus, the investigation on magnetic transition and transition temperature is significant in understanding magnetic properties and phenomena. In fact, due to the interaction between magnetic moment and lattice strain, which has a quantum mechanical origin and exists in all magnetic systems, magnetic materials can also demonstrate a magneto–elastic effect [3,4]. Since the moments of the rare-earth elements and the cobalt are antiparallel, the Laves-phase intermetallic RT_2_ compounds (R, rare-earth element; T, transition metal element, Fe/Co/Ni) are ferrimagnetic below Curie temperature (T_*C*_) [1,5]. Moreover, the Laves-phase RT_2_ compounds exhibit large magnetoelastic properties and have drawn much attention in the past few decades [6,7,8,9,10,11,12,13,14]. The Laves-phase RCo_2_ compounds, except the GdCo_2_, for which the Curie temperature (T_*C*_) is above 300 K, demonstrate paramagnetic–ferrimagnetic transition below 300 K, allowing the convenience of investigation from paramagnetic to ferrimagnetic states [10]. It has been extensively studied and well known that the distortion of the rare earth-based intermetallic compounds accompanying the change of the magnetic ordering (magnetic transition) is attributed to the interaction between the rare earth and the transition metals [8,10,11,12,14,15,16]. 

Previously, quite few magnetic materials, e.g., NiMnGa and La_1*-x*_Sr*_x_*MnO_3_ [17,18], were reported to exhibit structural transitions. In recent years, the investigations based on the high-resolution synchrotron X-ray diffraction technique proved experimentally that various ferromagnetic/ferrimagnetic materials undergo structural transition upon the ordering of magnetic moments, not as claimed with no change in the host crystal structure or symmetry [19,20]. Moreover, the crystal symmetry conforms to the easy magnetization axis (EMA): The crystals with EMA//[111] possess rhombohedral structural symmetry and the crystals with EMA//[001] possess tetragonal structural symmetry [21,22,23,24]. 

For the Tb_1-*x*_Dy_*x*_Co_2_ system, one endmember is TbCo_2_, which undergoes a paramagnetic–ferrimagnetic transition with temperature lowering, and its EMA is along [111]; the other endmember is DyCo_2_, which undergoes a paramagnetic–ferrimagnetic transition with temperature lowering, and its EMA is along [001]. The intermediate compositions (*x* = 0.3–0.8) undergo paramagnetic–ferrimagnetic–ferrimagnetic transitions [22]. Here in this work, exemplified with the Laves-phase Tb_1-*x*_Dy_*x*_Co_2_ compounds, the paramagnetic–ferrimagnetic and ferrimagnetic–ferrimagnetic transitions in view of structural transition, and the correlation between thermal expansion, magnetostriction, and the structural transition are demonstrated for the first time.

## 2. Materials and Methods 

The Tb_1*-x*_Dy*_x_*Co_2_ polycrystalline samples were prepared by arc melting method with the raw materials of Tb (99.9%), Dy (99.9%), and Co (99.9%) in argon atmosphere. The as-cast ingots were then annealed at 1000 °C for 24 hours. Synchrotron XRD with the X-ray wavelength of 0.65297 Å and a strain resolution of about 5 × 10*^−^*^4^ (BL15XU NIMS beam line of Spring-8, Hyogo, Japan) was employed to determine the crystal symmetries. The samples prepared for synchrotron XRD were powders, and sealed into quartz capillaries with a diameter of 0.3 mm. The capillary was rotated during the measurement to reduce the preferred orientation effect and to average the intensity. The thermal expansion and magnetostriction were measured using strain gauges (KFL-02-120-C1-11, Kyowa Electronic Instruments, Tokyo, Japan), with the measurement direction parallel to the applied magnetic field. In order to avoid the measurement error caused by the strain gauge itself with temperature variation, the thermal expansion was measured twice for each sample: Without magnetic field (0 Tesla) and with magnetic field (1 Tesla), and the thermal expansion versus temperature curves used for detecting the transition are obtained from the subtraction of the two. The temperature spectra of magnetization and magnetic susceptibility were measured using a superconducting quantum interference device (SQUID, Quantum Design, San Diego, USA). The applied magnetic field upon measurement of magnetization is 500 Oe, the applied magnetic field upon measurement of susceptibility is 50 Oe, and the measurement frequency is 133 Hz. 

## 3. Results and Discussion

### 3.1. The Mechanism of Thermal Expansion and Magnetostriction Based on Structual Transition and Domain Switching

Based on the feature of crystal symmetry breaking upon transitions, we schematically demonstrate the thermal expansion on the paramagnetic–ferrimagnetic transition, and the corresponding magnetostriction caused by domain switching in the ferrimagnetic phase. Figure 1a,b depicts the structural transition from cubic (*C*) to rhombohedral (*R*) phase and to tetragonal (*T*) phase, respectively. When the paramagnetic (*C*)–ferrimagnetic (*R*) transition occurs, the crystal with EMA//[111] elongates under the external field and then the measured expansion would be positive; correspondingly, the domain switching under the external field results in the EMA [111] aligning along the external field, thus yielding a positive magnetostriction (Figure 1c). When the paramagnetic (*C*)–ferrimagnetic (*T*) transition occurs, the crystal with EMA//[001] shrinks under the external field and then the measured expansion would be negative; correspondingly, the domain switching under the external field results in the EMA [001] aligning along the external field, thus yielding a negative magnetostriction (Figure 1d). The negative expansion and magnetostriction for the ferrimagnetic tetragonal structure (Figure 1b) originate from the c/a ratio of a tetragonal lattice being less than unity, which is caused by the ordered spin arrangement and the exchange repulsion between the nearest neighbors in the same (001) plane [25]. In brief, the lattice distortion upon transition would be reflected on the changes of thermal expansion and magnetostriction, both of which can be utilized to precisely probe the occurrence of the phase transition, as exemplified with Tb_1-*x*_Dy*_x_*Co_2_ (*x* = 0, 0.7, 1), as shown in the following:

### 3.2. Crystal Structure Symmetry and Phase Diagram

The structural evolution with temperature variation for TbCo_2_, Tb_0.3_Dy_0.7_Co_2_, and DyCo_2_ are shown in Figure 2. Figure 2a shows the phase diagram of the Tb_1*-x*_Dy*_x_*Co_2_ system [22], and Figure 2b–d shows the three reflections {222}, {440}, and {800} at different temperatures for TbCo_2_, Tb_0.3_Dy_0.7_Co_2_, and DyCo_2_, respectively. For TbCo_2_, at 265 K, there is no peak splitting in the three reflections, suggesting the cubic structural symmetry. At 195 K, there is no splitting in {800} and the intensity ratio of split peaks for {222} and {440} are 1:3 and 1:1, respectively, characterizing a rhombohedral structure. At 130 K, the rhombohedral splitting in {222} and {440} becomes more distinguishable. For Tb_0.3_Dy_0.7_Co_2_, at 200 K, there is no peak splitting in the three reflections, suggesting a cubic structural symmetry. At 125 K, there is no splitting in {800} and the intensity ratio of split peaks for {222} and {440} are 1:3 and 1:1, respectively, characterizing a rhombohedral structure. At 90 K, the splitting in {222} disappears, and the intensity ratio of split peaks for {440} and {800} are 1:2 and 2:1, respectively, characterizing a tetragonal structure. For DyCo_2_, at 200 K, there is no peak splitting in the three reflections, suggesting a cubic structural symmetry. At 130 K, there is no splitting in {222} and the intensity ratio of split peaks for {440} and {800} are 1:2 and 2:1, respectively, characterizing a tetragonal structure. At 60 K, the tetragonal splitting in {440} and {800} becomes more distinguishable. The corresponding lattice parameters are shown on the right side of XRD patterns.

### 3.3. Magnetic Properties, Thermal Expansion, and Magnetostriction

The thermal expansion and magnetostriction with temperature variation for TbCo_2_, Tb_0.3_Dy_0.7_Co_2_, and DyCo_2_ are shown in Figure 3. Figure 3a–c exhibits the temperature spectra of magnetization and magnetic susceptibility of TbCo_2_, Tb_0.3_Dy_0.7_Co_2_, and DyCo_2_, respectively, clearly indicating the phase transitions and transition temperatures for the three compositions. Furthermore, combined with the characteristic splitting of the three reflections, {222}, {440}, and {800}, from the synchrotron XRD patterns, the transition sequences are determined to be *C*–*R*, *C*–*R*–*T*, and *C*–*T*, respectively.

Figure 3d–f exhibits the thermal expansion of TbCo_2_, Tb_0.3_Dy_0.7_Co_2_, and DyCo_2_, respectively. Despite the polycrystalline state, the samples are applied with the magnetic field above T_*C*_ and the subtraction curve of the 1 Tesla and 0 Tesla are utilized to study the thermal expansion effect, so the non-magnetic thermal expansion (caused by temperature variation) is not considered here [11]. For TbCo_2_, a hump appears on the expansion curve at the Curie transition, and the expansion remains positive and keeps increasing on cooling. For Tb_0.3_Dy_0.7_Co_2_, two humps appear on the expansion curve, corresponding to the two transitions of *C*–*R* and *R*–*T*, and the expansion remains positive, but its value decreases on cooling below *R*–*T* transition temperature. For DyCo_2_, a peak appears at the Curie transition, and on cooling across T*_C_*, the expansion jumps abruptly from positive to negative, and then the absolute value of the expansion increases monotonously.

Figure 3g–i exhibits the temperature dependent magnetostriction curves of TbCo_2_, Tb_0.3_Dy_0.7_Co_2_, and DyCo_2_, respectively. For TbCo_2_, at 265 K (above T*_C_*), it delivers almost zero magnetostriction; at 195 K (below T*_C_*), the magnetostriction reaches 630 ppm; at 130 K (below T*_C_*), the magnetostriction reaches 2010 ppm. For Tb_0.3_Dy_0.7_Co_2_, at 200 K (above T*_C_*), it delivers almost zero magnetostriction. At 125 K (below T*_C_* and above the ferri–ferri transition temperature of 108 K), the magnetostriction reaches 640 ppm. At 90 K (below the ferri–ferri transition temperature of 108 K), the magnetostriction reaches 380 ppm, and it is obviously seen that the domain switching-resulted magnetostriction under the small field (1000 Oe) is negative [26,27]. For DyCo_2_, at 200 K (above T*_C_*), it delivers almost zero magnetostriction; at 130 K (below T*_C_*), the magnetostriction reaches −460 ppm; at 60 K (below T*_C_*), the magnetostriction reaches −1200 ppm.

According to the mechanism of magnetostriction described in Figure 1, the crystal in the tetragonal phase (c/a < 1) yields negative thermal expansion and negative magnetostriction, but it seems to contradict the experimental results in Figure 3e,h because both the thermal expansion and the saturated magnetostriction are positive. This is attributed to the two steps involved in the magnetization/magnetostriction process for polycrystalline samples, domain switching, and domain rotation [3,28]. For simplicity, Figure 1 depicts only the domain switching but not domain rotation. As for soft magnets, the magnetic field required for complete domain switching is usually not high (approximately 1000 Oe for Tb_0.3_Dy_0.7_Co_2_ at 90 K); if the field increases further, domain rotation would happen, and it is possible to alter the value of the finally measured magnetostriction. Therefore, it is understandable that when the magnetic field exceeds 1000 Oe, which is required for domain switching of Tb_0.3_Dy_0.7_Co_2_, the magnetostriction changes from negative to positive, and such a phenomenon is not rare in magnetostrictive materials [2,3,7,28]. 

Moreover, compared with the thermal expansion of TbCo_2_ and DyCo_2_, the thermal expansion of Tb_0.3_Dy_0.7_Co_2_ exhibits better temperature stability and is closer to zero. No doubt that the alloy with nearly zero thermal expansion can be obtained through precise composition controlling, and this finding would inspire the design of Permalloy-like functional materials [29,30]. 

To sum up, during the cooling process, the thermal expansion and magnetostriction exhibit similar temperature dependence, and the anomalous signal (hump/peak) appearing on the thermal expansion versus temperature curves precisely demarcates the transitions. 

### 3.4. Deduction of Thermal Expansion with Landau Polynomials

In order to better understand the thermal expansion upon phase transition, a Landau polynomial approach is employed to show the calculation of thermal expansion upon the paramagnetic (*C*)–ferrimagnetic (*R* or *T*) transition. Considering the magneto–elastic coupling effect, the free energy of a cubic ferrimagnetic crystal can be expressed as [5,21,31,32]: (1)$$\begin{array}{cc} F=F_{a}+F_{e}+F_{me} & =K_{1}\left(\alpha_{x}^{2}\alpha_{y}^{2}+\alpha_{y}^{2}\alpha_{z}^{2}+\alpha_{z}^{2}\alpha_{x}^{2}\right)+\frac{1}{2}c_{11}\left(e_{xx}^{2}+e_{yy}^{2}+e_{zz}^{2}\right)\\ & +c_{12}\left(e_{xx}e_{yy}+e_{yy}e_{zz}+e_{zz}e_{xx}\right)+\frac{1}{2}c_{44}\left(e_{xy}^{2}+e_{yz}^{2}+e_{zx}^{2}\right)\\ & +b_{1}M_{S}^{2}\left[e_{xx}\left(\alpha_{x}^{2}-\frac{1}{3}\right)+e_{yy}\left(\alpha_{y}^{2}-\frac{1}{3}\right)+e_{zz}\left(\alpha_{z}^{2}-\frac{1}{3}\right)\right]\\ & +b_{2}M_{S}^{2}\left(e_{xy}\alpha_{x}\alpha_{y}+e_{yz}\alpha_{y}\alpha_{z}+e_{zx}\alpha_{z}\alpha_{x}\right) \end{array}$$
where *F_a_* is the magnetic anisotropic energy, due to ordering of magnetic moments, *F_e_* is the elastic energy, *F_me_* is the magnetoelastic energy, *K_1_* is a temperature-dependent anisotropic constant, *α_x_*, *α_y_*, and *α_z_* are the direction cosines of *M_S_* with respect to the cubic axes, the *c_ij_* is the elastic modulus, and *b_1_* and *b_2_* are temperature-dependent magnetoelastic coupling coefficients. 

Minimizing the free energy with respect to all independent strains *e_ij_* (i.e., *∂**F*/*∂**e_ij_*= 0) yields a spontaneous lattice distortion *e_(ij)s_* to the initial cubic lattice. The matrix form of this spontaneous strain is:
(2)$$(e_{jj}{)}_{s}=(\begin{array}{ccc} -\frac{b_{1}}{c_{11}-c_{12}}\left(\alpha_{x}^{2}-\frac{1}{3}\right) & -\frac{b_{2}}{c_{44}}\alpha_{y}\alpha_{x} & -\frac{b_{2}}{c_{44}}\alpha_{z}\alpha_{x}\\ -\frac{b_{2}}{c_{44}}\alpha_{x}\alpha_{y} & -\frac{b_{1}}{c_{11}-c_{12}}\left(\alpha_{y}^{2}-\frac{1}{3}\right) & -\frac{b_{2}}{c_{44}}\alpha_{z}\alpha_{y}\\ -\frac{b_{2}}{c_{44}}\alpha_{x}\alpha_{z} & -\frac{b_{2}}{c_{44}}\alpha_{y}\alpha_{z} & -\frac{b_{1}}{c_{11}-c_{12}}\left(\alpha_{z}^{2}-\frac{1}{3}\right) \end{array})M_{s}^{2}$$

If the EMA is along [111] (ferrimagnetic state with a rhombohedral crystal symmetry), following the requirements of structure symmetry (α*_x_* = α*_y_* = α*_z_* = $1/\sqrt{3}$), the measured expansion along [111] is: (3)$$\varepsilon_{111}=-\frac{2\varphi }{3c_{44}}cos\alpha_{t}M_{s}^{2}$$
where *φ* is a temperature-independent constant and *α**_t_*is the angle between each of the two basis axes of the rhombohedral lattice. For qualitative comparison, only the temperature dependence of the primary order parameter *M_S_* is considered, and these parameters *φ*, *c*_44_, and *α**_t_* are taken constant. After plugging the square of *M_S_* (Figure 3a) into Equation (2), the thermal expansion along [111] can be calculated qualitatively, which is compared with the experimental curve, as shown in Figure 4a. 

If the EMA is along [001] (ferrimagnetic state with a tetragonal crystal symmetry), following the requirements of structure symmetry (*α**_x_* = *α**_y_* = 0 and *α**_z_* = 1), the measured expansion along [001] is: (4)$$\varepsilon_{001}=\frac{2\eta }{3\left(c_{12}-c_{11}\right)}\left(\frac{T}{T_{C}}-1\right)M_{s}^{2}$$
where *η* is a temperature-independent constant. Taking a similar approach as above, here, only the temperature dependence of the primary order parameter *M_S_* is considered, and these parameters *η*, *c*_11_, and *c*_12_ are taken constant. After plugging the square of *M_S_* (Figure 3c) into Equation (2), the thermal expansion along [001] can be calculated qualitatively, which is compared with the experimental curve, as shown in Figure 4b. 

The calculated thermal expansion curves for both TbCo_2_ and DyCo_2_ agree with the experimental curves. It should be noticed that for DyCo_2_, upon cooling and approaching T_*C*_, the expansion turns positive first and dramatically turns to negative when the temperature is below T_*C*_, and then its value gradually increases with further cooling. The peak on the experimental curve also appears on the calculated curve, which is clearly shown in the inset of Figure 4b. Such a peak is well described by Equation (4): While approaching T_*C*_, *M_S_* starts to increase from zero, due to the precursor effect in ferroic materials [33], resulting in a positive peak on the curve. The consistency of the calculated thermal expansion curves and the experimental ones verifies the theoretical analysis above. 

## 4. Conclusions

In conclusion, the paramagnetic–ferrimagnetic and ferrimagnetic–ferrimagnetic transitions of the Laves-phase alloys Tb_1*-x*_Dy*_x_*Co_2_ (*x* = 0, 0.7, 1) have been probed through the thermal expansion versus temperature curves and the magnetostriction. Because the interaction between magnetic moments and lattice strain has a quantum mechanical origin and exists in all magnetic systems [3,4], the magneto–elastic coupling effect authorizes the thermal expansion and magnetostriction to be fingerprints of ferrimagnetic transitions. In addition, it is further expected that not only the structural transition-involved ferrimagnetic transition, as investigated in the Laves-phase alloys in the present work, but also the ferrimagnetic transitions that have not been identified as structural transitions can be detected and investigated via measurements of elastic properties like thermal expansion and magnetostriction. 

## Figures and Tables

**Figure 1 materials-12-01755-f001:**
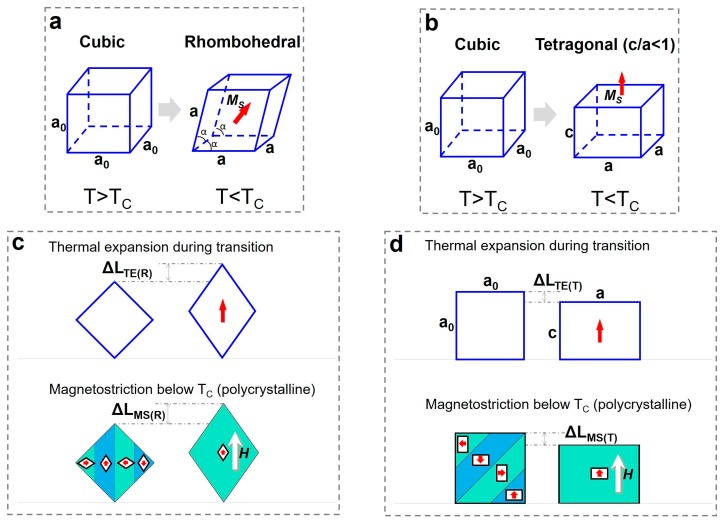
(Color online) Schematic illustrations of structural transition involved in paramagnetic (*C*)–ferrimagnetic (*R*) transition (**a**) and paramagnetic (*C*)–ferrimagnetic (*T*) transition (**b**); (**c**) shows the thermal expansion and magnetostriction for paramagnetic (*C*)–ferrimagnetic (*R*) transition and (**d**) shows the thermal expansion and magnetostriction for paramagnetic (*C*)–ferrimagnetic (*T*) transition.

**Figure 2 materials-12-01755-f002:**
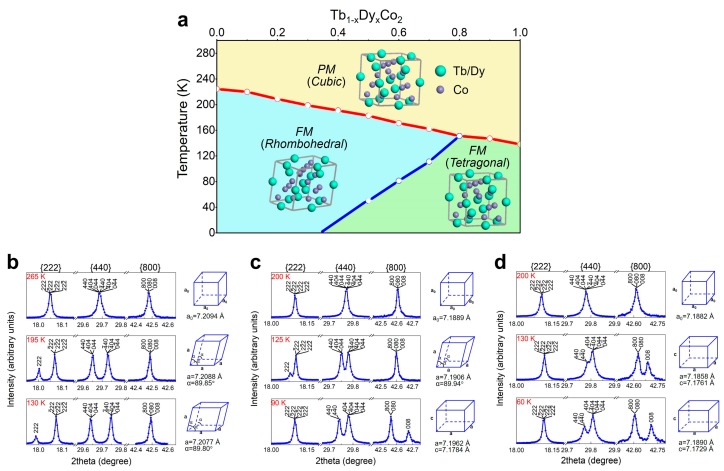
(Color online) (**a**) The phase diagram of the Tb_1*-x*_Dy*_x_*Co_2_ system; the reflections of {222}, {440}, and {800} from synchrotron XRD and the calculated lattice parameters for TbCo_2_ (**b**), Tb_0.3_Dy_0.7_Co_2_ (**c**), and DyCo_2_ (**d**) at different temperatures.

**Figure 3 materials-12-01755-f003:**
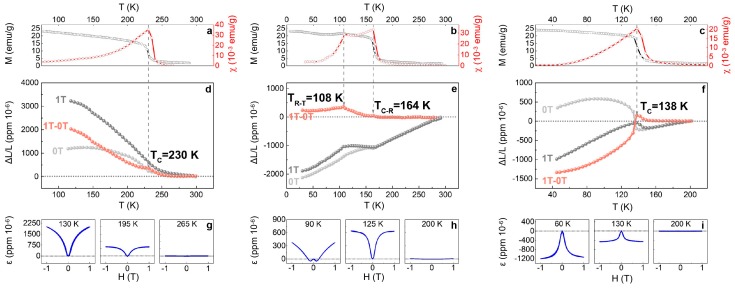
(Color online) (**a**–**c**) Temperature spectra of magnetization (M) and magnetic susceptibility (χ) of TbCo_2_, Tb_0__.3_Dy_0.7_Co_2_, and DyCo_2_; (**d**–**f**) the thermal expansion under 1 Tesla, 0 Tesla, and the subtraction of these two (1T–0T) of TbCo_2_, Tb_0__.3_Dy_0.7_Co_2_, and DyCo_2_; (**g**–**i**) the magnetostriction curves of TbCo_2_, Tb_0__.3_Dy_0.7_Co_2_, and DyCo_2_ at different temperatures.

**Figure 4 materials-12-01755-f004:**
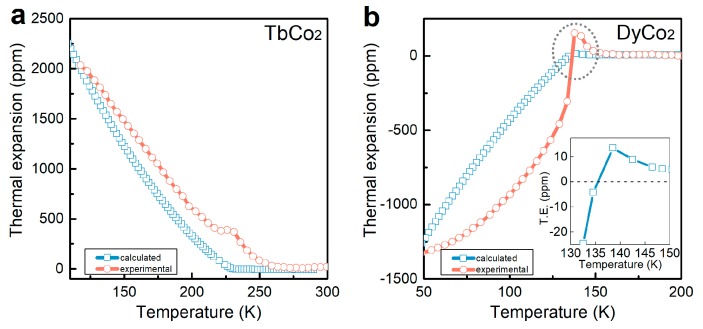
(Color online) The comparison between the calculated thermal expansion curve and the experimentally measured curve for TbCo_2_ (**a**) and DyCo_2_ (**b**).

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
