# Peer review of "Thermal Expansion and Magnetostriction of Laves-Phase Alloys: Fingerprints of Ferrimagnetic Phase Transitions"

_materials, 2019, doi:10.3390/ma12111755_

Round 1
Reviewer 1 Report
This manuscript is dedicated to studying the correlation between thermal expansion, magnitostriction and structural transitions in the TbDyCo2 system. In my opinion the manuscript is original and provides interesting results. I recommend publishing this manuscript after minor corrections given below.
1) In my opinion the text of manuscript should not include the expressions such as “we show” (lines 100, 121), “we employ” (line 172), because the manuscript is not the form of oral presentation. Please correct it.
2) Please write, that Tc is the Curie temperature (line 42), M is magnetization and χ is magnetic susceptibility (Fig.3, line 128).
3) It will be better for understanding, if you clarify in the introduction that “ferrimagnetic-ferrimagnetic transitions” are the transitions between compounds with different unit cell.
4) Please write what kind of device was used for the study of thermal expansion and magnetostriction (lines 66-67)?
5) The titles of the sections 3.1 and 3.2 are the same (”Crystal structure symmetry and phase diagram”). Please correct it.
6) Please increase the scale of Fig. 1, especially 1c, where the directions of red arrows are not visible.
7) Please make sure that you wrote correctly about "ferro-ferro transition" instead of “ferri-ferri transition” (lines 146, 147).
8) Are you sure, that the magnetostriction of the Tb0.3Dy0.7Co2 compound reaches 680 ppm at 90K? According to the Fig.3(c2) this magnetostriction value is about 380 ppm.
Author Response
Point 1: In my opinion the text of manuscript should not include the expressions such as “we show” (lines 100, 121), “we employ” (line 172), because the manuscript is not the form of oral presentation. Please correct it.
Response 1: Thanks for the reviewer’s comment. We have corrected these expressions accordingly in the revised version (Lines 110-111, 131-132, 182-183).
Point 2: Please write, that Tc is the Curie temperature (line 42), M is magnetization and χ is magnetic susceptibility (Fig.3, line 128).
Response 2: Thanks for the reviewer’s comment. We have made corresponding modifications in the revised version (Line 43, 138-139).
Point 3: It will be better for understanding, if you clarify in the introduction that “ferrimagnetic-ferrimagnetic transitions” are the transitions between compounds with different unit cell.
Response 3: Thanks for the reviewer’s comment. Some description sentences about the ferrimagnetic-ferrimagnetic transitions are added accordingly (Lines 38-40, 57-64).
Point 4: Please write what kind of device was used for the study of thermal expansion and magnetostriction (lines 66-67)?
Response 4: Thanks for the reviewer’s suggestion. The necessary information about the device is provided in the revised version (Lines 73-74).
Point 5: The titles of the sections 3.1 and 3.2 are the same (”Crystal structure symmetry and phase diagram”). Please correct it.
Response 5: Thanks for the reviewer’s reminding. We have corrected the title of the section 3.1 accordingly in the revised version (Lines 84-85).
Point 6: Please increase the scale of Fig. 1, especially 1c, where the directions of red arrows are not visible.
Response 6: Thanks for the reviewer’s suggestion. The Fig.1 is enlarged in the revised version.
Point 7: Please make sure that you wrote correctly about "ferro-ferro transition" instead of “ferri-ferri transition” (lines 146, 147).
Response 7: Thanks for the reviewer’s comment. The typos are corrected in the revised version (Lines 156-157).
Point 8: Are you sure, that the magnetostriction of the Tb0.3Dy0.7Co2 compound reaches 680 ppm at 90K? According to the Fig.3(c2) this magnetostriction value is about 380 ppm.
Response 8: Thanks for the reviewer’s careful review. The typo is corrected in the revised version (Line 158).
Reviewer 2 Report
The Author presents work related with the pseudo-binary Laves-phase Tb1-xDyxCo2 system (x=0.0, 0.7 and 1.0), where the thermal expansion and magnetostriction can probe the ferromagnetic transitions from: a) Cubic to Rhombohedral phase (in TbCo2), b) Cubic to Tetragonal phase (in DyCo2), c) Cubic to Rhombohedral and then to Tetragonal phase (in Tb0.3Dy0.7Co2). It is nice and complex work (with different techniques) combined with some mathematical approach. Obtained results of their studies agree (with some error) with calculated thermal expansion.
However, I have question to Authors:
- In section 3.3 Authors mention about domain – they rotation or switching. Did they try to observe some magnetic domains by e.g. magnetooptical techniques or They conclude this information from hysteresis loops?
I also suggest:
- to check English grammar in the text,
- for better reading it could be nice to have slightly shorter sentences: please have a look e.g. on line 103 and next ones.
Author Response
Point 1: In section 3.3 Authors mention about domain – they rotation or switching. Did they try to observe some magnetic domains by e.g. magneto-optical techniques or they conclude this information from hysteresis loops?
Response 1: Thanks for the reviewer’s comment. Actually, we did not observe the domain switching in the present work. This is concluded from the shape of the magnetostriction curves. The W-shaped magnetostriction curves have been observed in other magnetic systems and well understandable in view of domain rotation/switching. In the revised version, more references are cited to make it clear (Line 159).
Point 2: I also suggest: (a). to check English grammar in the text; (b). for better reading it could be nice to have slightly shorter sentences: please have a look e.g. on line 103 and next ones.
Response 2: Thanks for the reviewer’s suggestions. We checked English grammar and made corresponding corrections in the revised version (Lines 90-97, 113-125, 145-151, 155-159, 191-194).
Reviewer 3 Report
The paper by Zhou et al. reports an original investigation of the magnetostrictive effect in Tb1-xDyxCo2 system (x=0.0, 0.7 and 1.0). The paper is sound and well written. I would reccommend some minor changes before publication:
1) Materials and methods: please give some more details on the synthesis (for example, synthesis tempearture, holding toime, etc...);
2) Materials and methods: please give some more details on the strain gauges used for the measure of thermal expansion and magnetostriction;
3) A paragraph concerning domain rotation at the end of section 3.1 would be beneficial for the following discussion of the data;
4) Section 3.4: formula 1 is valid only for a cubic lattice. Please specify this in the text;
5) line 197: alpha t: t is a subscript
6) In Figure 4, the calculated thermal expansion is reported and compared to the experimental one. What are the values of the parameters of equations 3 and 4 used to the calculations? How were they obtained?
Author Response
Point 1: Materials and methods: please give some more details on the synthesis (for example, synthesis temperature, holding time, etc...).
Response 1: Thanks for the reviewer’s comment. The necessary information about the synthesis is provided in the revised version (Lines 67-68).
Point 2: Materials and methods: please give some more details on the strain gauges used for the measure of thermal expansion and magnetostriction.
Response 2: Thanks for the reviewer’s comment. The information about the strain gauges is provided in the revised version (Lines 73-74).
Point 3: A paragraph concerning domain rotation at the end of section 3.1 would be beneficial for the following discussion of the data.
Response 3: Thanks for the reviewer’s suggestion. Some sentences are modified accordingly in the revised version (Lines 90-97).
Point 4: Section 3.4: formula 1 is valid only for a cubic lattice. Please specify this in the text.
Response 4: Thanks for the reviewer’s comment. This is specified in the revised version (Line 185).
Point 5: line 197: alpha t: t is a subscript.
Response 5: Thanks for the reviewer’s comment. The typo has been corrected accordingly (Line 207).
Point 6: In Figure 4, the calculated thermal expansion is reported and compared to the experimental one. What are the values of the parameters of equations 3 and 4 used to the calculations? How were they obtained?
Response 6: Thanks for the careful review. To be honest, in the present study, we did not obtain the values for the parameters (φ, c44 and αt for Equ.3; η, c11 and c12 for Equ.4) experimentally. Motivated by the qualitative comparison, the values of the parameters for the calculated thermal expansion were fixed to satisfy the following condition: the calculated thermal expansion is close to the experimental one. We believe that, lacking of the experimental values of these parameters is the reason for the deviation of the calculated expansion from the experimental one. In the revised version, some sentences are added to make this point clear (Lines 208-211, 221-223).